# Dignity-based care and infertility treatment: A qualitative study

**Zahra Kiani[1], Masoumeh Simbar[2]\*, Soheila Nazarpour[3], Vida Ghasemi[4]**

**1** Midwifery and Reproductive Health Research Center, Shahid Beheshti University of Medical Sciences, Tehran, Iran, **2** Department of Midwifery and Reproductive Health, Midwifery and Reproductive Health Research Center, School of Nursing and Midwifery, Shahid Beheshti University of Medical Sciences, Tehran, Iran, **3** Department of Midwifery, Chalous Branch, Islamic Azad University, Chalous, Iran, **4** Department of Midwifery, Asadabad School of Medical Sciences, Asadabad, Iran

\* msimbar@gmail.com

## Abstract

### Introduction

The treatment of infertile people is generally time-consuming and requires frequent and long-term visits and providing dignity-based services. Due to the different perceptions and experiences of people and the lack of a specific study to explain the concept of dignity-based care, this study aimed to explain the concept and dimensions of dignity-based care in infertility treatment services.

### Methods

This was a qualitative study with a conventional content analysis approach. Fifty participants (20 infertile women, 16 infertile men, and 14 key informants) were recruited using a purposive sampling method from an educational center of Mazandaran University of Medical Sciences and a private infertility center in Mazandaran –Iran in 2023. Sampling was continued until data saturation. Data were collected using in-depth and semi-structured individual interviews. The data were also analyzed using the conventional content analysis method and the steps suggested by Grandheim and Lundman. Also, Lincoln and Guba's criteria were used to check the trustworthiness of the data.

### Results

The content analysis demonstrated 43 codes in 11 sub-categories and 4 categories. These categories are "conserving dignity in providing care", "making the information accessible and obtaining informed consent for care procedure", "providing professional care and standard services", and considering cultural and social aspects of infertility. The categories were used to explain the concept and dimensions of

**Data availability statement:** There are ethical restrictions on sharing minimal data publicly for this study, because the data contain potentially identifiable participant information. Data are available upon request from the Ethics Committee of Shahid Beheshti University of Medical Sciences (https://ethics.research. ac.ir/PortalCommittee.php?code=IR.SBMU. PHARMACY.REC) for researchers who meet the criteria for access to confidential data.

**Funding:** The author(s) received no specific funding for this work.

**Competing interests:** The authors have declared that no competing interests exist.

dignity-based care in infertility treatment services which are showing the multidimensional aspects of this concept.

## Conclusion

Dignity-based care in infertility treatment services means "conserving dignity in providing care services; making the information accessible and obtaining informed consent for care procedure; providing professional care and standard services; and considering cultural and social aspects of infertility." This concept can be used in future policy-making and planning, and appropriate support should be taken into account to improve the quality of infertility treatment services.

### Introduction

Infertility is an important problem in reproductive health. It is defined as not being able to conceive after one year or longer of unprotected sex [1]. The trend of infertility in the world is growing in recent years, and according to the latest report of the World Health Organization (WHO), in 2023, the prevalence of periodic and lifetime infertility was 12.6% and 17.5%, respectively [2].

Infertility mostly hurts various dimensions of health and causes a decrease in the couples' quality of life [3]. Infertile couples may experience some psychosocial disorders such as anxiety, depression, stress, low self-confidence, social isolation, despair, social stigma, and so on [4,5].

There are various methods to treat infertility that require frequent visits and high costs of associated treatment [6]. Infertile couples do not disclose their infertility, especially in developing countries, and for this reason, sometimes they do not seek treatment [7]. So, it is critical to keep human values in consideration, the confidentiality of information, and providing dignity-based care to them [3,4].

Conserving dignity (respect, privacy, effective communication) in providing care services is an approach that is designed based on ethical principles, prioritizes people's preferences, and is based on evidence-based practices [8]. Providing respectful care with empathy, support, and by empowering and improving the client's self-confidence are the main features of dignity-based care [9]. In recent years, promoting dignity-based care and considering the cultural, social, and psychological aspects of care are emphasized by national and international organizations [10].

Universal access to sexual and reproductive health services, including infertility treatment services, is accentuated in the sustainable development goals [11]. There is a delay in seeking treatment due to the limited access and unaffordability of assisted reproductive technology (ART) in most parts of the world [12]. Besides, considering the specific mental conditions of infertile people, preserving human values, paying attention to their expectations, and providing dignity- oriented care in infertility treatment are essential [13–15]. However, there is no precise definition of dignity-based care in infertility treatment, while addressing it can be very effective on the consequences of infertility and its treatment [16] as well as on the planning for special dignity- oriented services [17].

In a qualitative study on infertile women, Kiani and colleagues emphasized that it is important to follow ethical principles in providing infertility services to reduce the clients' anxieties [3]. Bezad et al. also stated that universal coverage and access to infertility care services is a right and not an option, and providing infertility services along with preserving dignity is an important component of providing infertility services and it should be mentioned and defined as the rights of infertile couples by the health policymakers and program managers [14]. Furthermore, Gipson et al., in their review article concluded that infertility is one of the components of sexual and reproductive health and rights that is usually ignored in planning for providing dignity-based services [18].

Although there is a general agreement for dignity-based services, it is ignored in many services and the client's needs are not taken into account [19,20]. The purpose of qualitative research is to understand and explain different people's perceptions and experiences of a phenomenon that usually differ from one person to another [21]. Therefore, this qualitative study aimed to explain the concept and dimensions of dignity-based care in infertility treatment services.

## Materials and methods

### Design of the study

This was a qualitative study with a conventional content analysis approach.

### The participants

In this qualitative study, infertile men and women and key informants (including specialists in reproductive health, medical ethics, a gynecologist with infertility fellowship, midwives with an experience in infertility services, the health policy maker, as well as an anthropologist, a clinical psychologist and a psychiatrist who had experience with infertile couples. The inclusion criteria for participating in the study include women and men with a one- year history of infertility diagnosed by a gynecologist, with permanent marriage, no adoption, no chronic disease, and no psychiatric disorders. The key informants had at least two years of experience in infertility services, with a minimum Bachelor's degree.

### Sampling

Purposive sampling method was used for the participants' recruitment and it was continued until data saturation. Data saturation meant not creating new codes in the interviews, and after saturation, three more interviews were conducted to ensure its accuracy. All eligible participants accepted to be interviewed and no one refused to participate.

### Setting

The places of the study were an educational hospital affiliated with Mazandaran University of Medical Sciences as well as a private infertility center in Sari City in April 1, 2023 to December 14, 2023. The interview was performed in a private room in the infertility treatment center. Key information was obtained in their work office.

### The tool for data collection

The researcher used various methods for data collection including semi-structured interviews, observation, audio recording, writing them down, taking field notes, and document review. The in-depth face-to-face interview began with semi-structured questions such as

Were you satisfied with the services provided at the infertility centers? Do you think these services were within the framework of ethical standards for you? Can you give me an example? Was there anything where you felt that ethics were not observed in the provision of services? Was there any case where you felt that your rights were violated? What do you think dignity means in your care? Do you feel that you received dignified care? All interviews were performed and

recorded with the written consent of the participants. The average duration of the interview was around 45 minutes. All people were interviewed once only.

## Trustworthiness

The trustworthiness of data was analyzed using four criteria suggested by Lincoln and Goba (1994) including credibility, dependability, transferability, and confirmability [22]. Besides, the authenticity of the data was assessed as the fifth criterion that is suggested by Polit and Beck [23].

**Credibility.** The creditability of data was increased through methods such as (1) prolonged engagement and persistent observation; by spending adequate time for data collection and gaining the trust of the participants, (2) member checking: by asking participants to review and provide feedback on the accuracy of the extracted codes, and to ensure the similarity of the findings with the participants' experiences, (3) peer debriefing: by asking two other researchers to review the research process and finding to ensure objectivity and accuracy of data, (4)Triangulation; using multiple methods and source for data collection such as in-depth individual interview and observation and the literature review, and (5) Improving interview skills: the interviewer (ZK, 32- years-old women, PhD of reproductive health, assistant professor of midwifery and reproductive health research center and infertility researcher) improved her skills by participating in the interview and qualitative content analysis courses.

**Dependability.** The dependability of the data was increased by asking all the participants questions in the same field, providing a detailed description of the research methods, independent audit and coding of the interviews by two reproductive health outside the research team, and also the code-recode method.

**Transferability.** Transferability of the data was improved by leaving aside the thoughts of the researcher, providing details about the context and the procedure of the study, and agreement of two reproductive health experts on the formation of codes and categories in data analysis.

**Confirmability.** It was increased by considering maximum diversity in terms of age, education, employment status, type of treatment, and duration of infertility, and providing a rich description of the qualitative method.

**Authenticity.** It was improved by conveying the feeling tone of infertile couples' lives by providing vicarious experience, feeling, language, and context of those lives.

## Data analysis

Conventional content analysis was performed based on the criteria proposed by Graneheim and Lundman [22]. After transcribing the recorded interviews, the text was carefully reviewed by the researcher several times to achieve an accurate understanding of the interview contents. Then, the text was divided into meaning units, meaning units were condensed while preserving the meaning and labeled with codes. Similar codes were then categorized into subcategories, and the subcategories were classified into a category based on common properties. The latent content of the similar categories was finally formulated as a theme. We transcribed all interviews with women, men, and key informants and put similar codes together to arrive at the concept of a dignity-centered care unit.

## Ethics

The study was approved by the ethical committee of Shahid Beheshti University of Medical Sciences (IR.SBMU. PHARMACY.REC.1401.287). We confirm that all methods were performed in accordance with the relevant guidelines and regulations. Before conducting the interviews, the researcher briefed the participants on the objectives of the study and ensured the confidentiality of their information and the voluntary type of participation. Informed written and verbal consent was also obtained from all participants for the interview and audio recording of the interview.

# Results

Twenty infertile women and 16 infertile men and 14 key informants participated in the study. The demographic characteristics of the participants and the key informants are shown in Tables 1 and 2, respectively.

**Table 1. Demographic characteristics of women and men participating in the study "explaining the concept of dignity-based care in infertility treatment services".**

| Characteristics | | Mean (Women) | Standard deviation (Women) | Mean (Men) | Standard deviation (Men) |
|---|---|---|---|---|---|
| Age (years) | | 32.16 | 9.68 | 34.58 | 8.79 |
| Duration of marriage | | 11.41 | 6.72 | 15.32 | 7.32 |
| Duration of infertility | | 7.82 | 7.95 | 8.05 | 6.45 |
| | | Number | Percent (%) | Number | Percent (%) |
| Education | Elementary | 4 | 20.0 | 2 | 12.5 |
| | High school | 4 | 20.0 | 3 | 18.75 |
| | Diploma | 4 | 20.0 | 3 | 18.75 |
| | Bachelor's degree | 3 | 15.0 | 3 | 18.75 |
| | Master's degree | 3 | 15.0 | 3 | 18.75 |
| | Doctorate | 2 | 10.0 | 2 | 12.5 |
| Residence | City | 13 | 65.0 | 9 | 56.25 |
| | Village | 7 | 35.0 | 7 | 43.75 |
| The economic situation | Low | 6 | 30.0 | 4 | 25.0 |
| | Medium | 8 | 40.0 | 8 | 50.0 |
| | High | 6 | 30.0 | 4 | 25.0 |
| Type of infertility | Primary | 12 | 60.0 | 10 | 62.5 |
| | Secondary | 8 | 40.0 | 6 | 37.5 |
| Type of treatment | Medication | 5 | 25.0 | 4 | 25.0 |
| | IUI | 5 | 25.0 | 4 | 25.0 |
| | IVF | 5 | 25.0 | 4 | 25.0 |
| | Donor eggs | 5 | 25.0 | 4 | 25.0 |

**Table 2. The characteristics of women and men participating in the study "explaining the concept of dignity-based care in infertility treatment services".**

| Profession | Age (years) | Work experience (years) | Education |
|---|---|---|---|
| Medical ethics | 52 | 29 | PhD |
| Medical ethics | 40 | 10 | PhD |
| Health policy maker | 43 | 13 | PhD |
| Obstetrician gynecologist with infertility fellowship | 55 | 22 | MD |
| Obstetrician gynecologist with infertility fellowship | 46 | 15 | MD |
| Anthropologist | 48 | 20 | PhD |
| Reproductive health specialist | 54 | 21 | PhD |
| Reproductive health specialist | 42 | 10 | PhD |
| Clinical Psychologist | 40 | 11 | PhD |
| Clinical Psychologist | 45 | 20 | BS |
| Psychiatrist | 45 | 17 | MD |
| Midwife | 44 | 16 | BS |
| Midwife | 37 | 12 | MS |
| Midwife | 31 | 7 | MS |

The content analysis of the data demonstrated 43 codes, 11 sub-categories, and 4 categories (Table 3).

## Conserving dignity in providing care

This theme consists of three categories, respect, privacy, and effective communication.

**Respect.** All of the participants emphasized providing respectful care as an important approach to providing dignity-based care, through performing procedures such as Introducing the clinic personnel and their tasks, providing patient-friendly clinic environments, providing proper and on-time services, providing welfare facilities, and avoiding discrimination in providing the services.

*"When I arrived in the clinic, a lady in charge introduced herself to us and explained about the services for infertile couples in the clinic. I felt relaxed and I thought that she respects me and my husband" (32-year-old woman, housewife, 8-year history of primary infertility).*

*"Sometimes we wait a lot here. I wish the timings were more precise. This is a way and a meaning for providing respectful care" (45-year-old man, 15-year history of primary infertility).*

*"I came to the center from the nearby cities for treatment and it is difficult for me to travel between the cities. I wish the center had a place for us to stay, where we would reserve the place in advance so that we don't have to worry about our accommodation. In this way, one feels more respected" (38-year-old woman, housewife, 10-year history of secondary infertility).*

**Privacy.** The majority of participants stated that providing privacy in the care procedure is a critical item for dignity-based care. They mentioned providing a private room for counseling, physical exams, and care procedures; the absence of unnecessary attendees during counseling and care procedures; and considering the confidentiality of the information.

*"My wife and I expect that the infertility center keeps our information a secret. Sometimes my mother insists on coming with us, but I don't want her to come with us, and I like that even when she comes with us, I expect the doctor allows no one to be there except me and my wife "(37- years-old man, 5-years history of primary infertility).*

*"I don't like that our personal information such as the cause of our infertility, the treatment procedures, and the treatment repetition, to be revealed to anyone; as these are from our private issues" (32-year-old woman, housewife, 12-year history of infertility).*

**Effective communication.** All participants mentioned effective verbal and non-verbal communication in providing couple's counseling procedures.

*"Couples often say that providing couple's counseling with effective communication help in improving marital relationship, and they suppose that their dignity is being preserved" (Clinical Psychologist, 20 years of work experience).*

*"It is very important to me that when I talk to the doctor or the treatment personnel, they listen to me carefully and sometimes confirm my words by shaking their head or with their eyes" (26- year-old woman, housewife, 5-year history of primary infertility).*

## Access to information and considering informed consent

This theme is extracted as a significant dimension of the concept of dignity-based care in infertility services which means making awareness, freedom of choice, and participatory decision-making.

**Table 3. The codes, subcategories and categories of the concept dignity-based care in infertility treatment services.**

| Categories | Sub-Categories | Codes |
|---|---|---|
| Conserving dignity in providing care | Respect | Introducing of the clinic personnel and their tasks |
| | | Providing patient-friendly clinic environments |
| | | Providing proper and on time services |
| | | Providing welfare facilities |
| | | Avoiding discrimination in providing the services |
| | Privacy | Providing private room for counseling, physical exams and care procedures |
| | | Absence of unnecessary attendees during counseling and care procedures |
| | | Considering confidentiality of the information |
| | Effective communication | Effective verbal communication |
| | | Effective non-verbal communication |
| | | Providing couples' counseling services |
| Access to information and considering informed consent | Making Awareness | Providing the necessary information about the infertility problem to the couples |
| | | Explaining about the different treatment methods |
| | | Providing information about the procedure of the treatment |
| | | Informing the couples about the success rate |
| | | Providing information about the costs |
| | | Awareness about treatment repeatability |
| | Freedom to choose | Avoiding coercion for treatment time |
| | | Giving adequate information to make correct decision |
| | | Giving freedom to choose the treatment team |
| | Participatory decision making | Considering the couples' needs, expectations and demands |
| | | Obtaining permission to perform care procedures |
| | | Providing a couple-friendly atmosphere for making decisions about treatment |

*(Continued)*

**Table 3.** (Continued)

| Categories | Sub-Categories | Codes |
|---|---|---|
| Providing professional care and standard services | Professional care | Providing a team-work services |
| | | Providing mental care |
| | | The availability of the treatment team |
| | | Performing care procedures based on the protocols |
| | Standard services | Avoiding unnecessary interventions |
| | | Providing on time services |
| | | Increasing the number of public infertility centers |
| | | Providing adequate infertility specialists |
| | | Providing appropriate insurance coverage |
| | | Government financial support |
| Considering the cultural and social aspects of infertility | Attention to the common beliefs of society | Children as the life support |
| | | Difficulty in acceptance of couples' life without kids |
| | | Considering infertility as a female issue |
| | | Sterility stigmatization |
| | Modification of misbeliefs and attitudes | Considering the family education |
| | | The Community education |
| | | Involving national media |
| | recruiting the eligible personnel | Providing native treatment team |
| | | Treatment by a same-sex care provider |
| | | Training of non-native treatment team |

**Making awareness.** The majority of participants mentioned making awareness as an important factor in providing dignity-based care in infertility services. This category includes the codes such as providing the necessary information about the infertility problem to the couples; explaining the different treatment methods; providing information about the procedure of the treatment; informing the couples about the success rate; and providing information about the costs.

*"It is very important to be aware of different methods of infertility treatment and the treatment personnel should explain this to us" (33-year-old woman, housewife, 15-year history of primary infertility).*

*"I would like the doctor to explain to me about the chance of the treatment for example by IVF" (40-year-old woman, employed, 7-year history of secondary infertility).*

*"Some doctors use English words, for example, they say IVF or IUCS? I wish they could tell us in simple language what these are?" (33 years old man, 6-years history of primary infertility).*

*"I think it is very important to know how likely different methods could be successful in treating infertility and to have this information at our disposal" (45-year-old woman, housewife, 20-year history of primary* infertility).

**Freedom to choose.** The majority of participants emphasized avoidance of coercion for treatment time; giving adequate information to make correct decisions; and giving freedom to choose the treatment team as the means for providing dignity-based care services.

*"I like to be informed about all the procedures that should be provided for me with details and take my consent before performing the procedure." (37-year-old man, 5-years history of primary infertility).*

*"I am an employee and I came here from another province for treatment and it was difficult for me to take time off. I would like that they ask me when it is possible for me, rather than making an appointment by themselves" (45-year-old woman, Employed, 12-year history of primary infertility).*

*"I can't communicate with some doctors. I wish it was possible to choose the doctor and treatment team ourselves" (33-year-old woman, housewife, 16-year history of primary infertility).*

**Participatory decision making.** The participants mentioned considering couples' needs, expectations, and demands; obtaining permission to perform any care procedures, and providing a couple-friendly atmosphere for making decisions about treatment as the meaning of participatory decision-making.

*"I think we should have a role in making decisions about any care that we receive, and we should be able to talk about that easily" (40-year-old woman, employed, 9-years history of primary infertility).*

*"In my opinion, we should be consulted and informed before any care procedure and take our permission first" (33 years old man, 6-year history of primary infertility).*

*"The doctor and other personnel of infertility centers should create an appropriate atmosphere where people can express their wishes about treatment (psychiatrist, 17 years of work experience).*

**Providing professional care and standard services**

Providing professional care and standard services was explained as the most important meaning of providing dignity-based care services.

**Professional care.** Many participants described providing professional care as a critical aspect of dignity-based care services that can be demonstrated by providing a teamwork service; availability of the treatment team; providing mental care; and performing care procedures based on the protocols.

*"The treatment personnel should be a team that should also be available to answer the questions, solve the problems and concerns" (30-year-old man, 3-years history of primary infertility).*

*"One of the important aspects of infertile couples' care is attention to their psychological changes that should be considered in providing the professional care" (medical ethics specialist, 29-years history of work experience).*

**Standard services.** Several participants talked about providing standard services as a dignity-based care service which means avoiding unnecessary interventions; providing on-time services; increasing the number of public infertility centers; providing adequate infertility specialists; providing appropriate insurance coverage; and providing financial support by the government.

*"The training of the specialized infertility treatment staff is one of the important requirements in providing dignity-oriented infertility services" (Infertility fellow, 22 years of work experience).*

*"Sometimes I feel that the lack of financial support and appropriate insurance for infertility treatment makes us deprived of standard care treatment (28-year-old woman, working, 5 years of primary infertility history).*

*"Considering the recent policy of rejuvenating the population in the country, one of whose strategies is to help with the treatment of infertile couples, it is necessary to provide public infertility treatment services, with sufficient teams of specialists and experienced personnel and with an appropriate financial support or insurance." (A reproductive Health specialist with, 21- years of work experience).*

### Considering the cultural and social aspects of infertility

Considering the cultural and social aspects of infertility is an important dimension for providing dignity-based services. This approach can be achieved by a few strategies such as attention to the common beliefs of society; modification of misbeliefs and attitudes; and recruiting eligible personnel.

**Attention to the common beliefs of society.** The majority of participants stressed some common misbeliefs in society that are necessary to be considered in dignity-based care services, such as considering children as the support of life; or difficulty in accepting couples' life without kids; considering infertility as a female's problem; and sterility stigmatization.

*"In traditional societies, a woman's identity is completed when she has children" (anthropologist, 20 years of work experience).*

*"Some people in our society still consider the child as a life supporter." (A reproductive Health specialist with, 21-years of work experience).*

*It is difficult for the community to accept child-free families. These couples may feel stigmatized. (A reproductive Health specialist with, 10-years of work experience).*

*"Most people think that the problem of infertility is only related to women" (45-year-old woman, housewife, 12-year history of secondary infertility).*

**Modification of misbeliefs and attitudes.** All Participants emphasized the critical role of a dignity-based service in modifying misbeliefs. They told that family and community education, as well as national media involvement for this modification, are necessary.

*"I wish families would be taught in the infertility center that my husband and I are undergoing treatment and there is no need for their interference" (33-year-old woman, employed, 7-year history of primary infertility).*

*"One of the important issues in the field of infertility care is educating the family so that their beliefs and attitudes can be corrected" (Reproductive health specialist, 10 years of work experience).*

*"In my opinion, the dignity-based infertility treatment centers should attempt to involve national media for correcting the misbeliefs about infertility in the community" (anthropologist, 20 years of work experience).*

**Recruiting eligible personnel.** The majority of personnel mentioned their preference in providing care by a native treatment team; female service providers provide services to women and men provide services to men and by trained nonnative personnel.

*"When the treatment provider is from our local area, I feel it is easier to understand our words and concerns" (27-year-old woman, housewife, 4-year history of secondary infertility).*

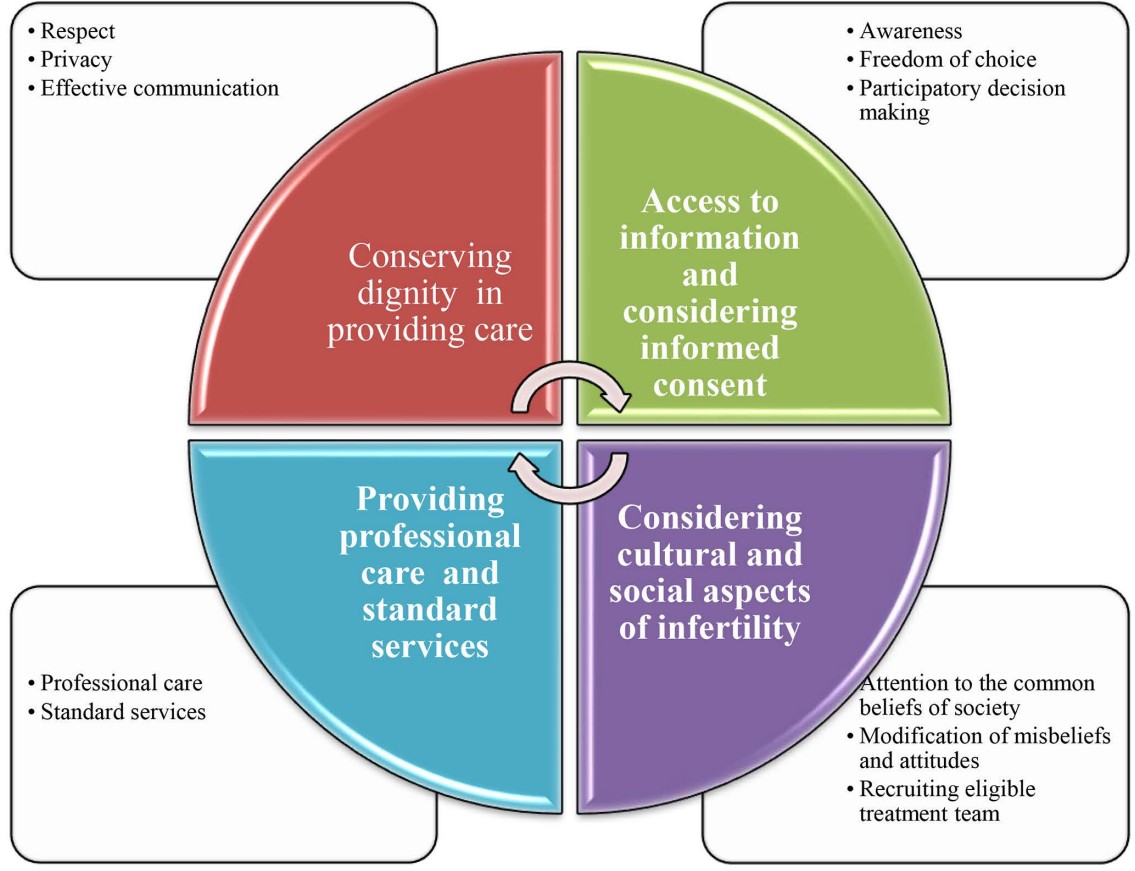

**Fig 1. The concept of dignity-based care in infertility treatment services.**

*"The native treatment personnel can understand well how to look at us as an infertile couple in the family and the community (36-year-old woman, housewife, 13-year history of primary infertility).*

### The concept of dignity-based care in infertility treatment services

The concept of "Dignity-based care in infertility treatment services" is conserving dignity (respect, privacy, effective communication) in providing care services; making the information accessible and obtaining informed consent for care procedures; providing professional care and standard services; with considering cultural and social aspects of infertility (Fig 1).

### Discussion

This was the first study to explain the concept of dignity-based care in infertility treatment services. The study demonstrated the concept with four dimensions including conserving dignity in providing care; making the information accessible and obtaining informed consent for care procedure; providing professional care and standard services; with considering cultural and social aspects of infertility. Immanuel Kant was one of the first people who defined dignity as a quality belonging to all sentient human beings because of their capacity for rationality and moral freedom [24], but generally, the concept of dignity is complex and ambiguous [25]. According to Kant's principles, if a person is deprived of the capacity for rational and moral self-determination (e.g., in a Persistent vegetative state, in a coma, etc.), they lack dignity while Aquinas

considers dignity to be the fundamental perfection of man that distinguishes him from other beings (things), therefore, as long as a human being exists, they have a dignity proper to themselves, regardless of changing qualities. Hence, this can be called the existential dignity [26]. Nowadays, many researchers are trying to explain the concept and dimensions of dignity in different areas of healthcare system as it is known as an Inalienable right of patients [27,28].

Conserving dignity in providing care was found as a dimension of dignity-based care in infertility treatment services that can be achieved by providing respectful care, keeping the clients' privacy, and making effective communication between the providers and the clients. Conserving human dignity is at the center of moral theories [29] and it is an intrinsic value that every human being seeks and likes to experience [30]. The options for infertility treatment range from providing simple counseling to invasive medical interventions, and the infertile clients must provide personal and family information including a detailed history of their physical, mental, and reproductive health condition, as well as the results of biochemical-, Genetic- and screening tests and also imaging. They may be recommended for accepting sperm or egg donation, or surrogacy. Therefore, providing respectful care and counseling, with keeping the clients' privacy and confidentiality of the information is critical [31]. On the other hand, they generally wish their information about the cause of infertility, the course of treatment, and the number of treatment cycles to be kept confidential [32], because they feel that disclosing the information increase the family and social pressure on them and also their privacy will be in threat [33]. Since pregnancy is a critical aim in marital life, infertile couples experience extensive changes in their health especially mental health [34]. Therefore, effective communication between infertile couples with the treatment provider is essential to conserve the clients' dignity [15].

The results demonstrated that making the information accessible for the clients and obtaining informed consent is essential for dignity-based care in infertility treatment services. This aspect of dignity care is achievable by providing the necessary information about infertility, the different treatment methods, the procedure of the treatment, the success rate, and also the costs. In a qualitative study by Kiani et al., the majority of participants stated that they would like to receive detailed information about the different methods of treatments, the repeatability of the treatments, and their rate of success [3]. The Council for International Organizations of Medical Sciences (CIOMS) defines "informed consent" as (1) receiving the necessary information to make an informed choice, (2) understanding the information, and (3) making a voluntary decision [35]. Besides, in the Belmont Report by the National Commission for the Protection of Human Subjects of Biomedical and Behavioral Research, the importance of providing information, obtaining informed consent, and making independent care decisions was emphasized [36]. Failure to give necessary information to the clients reduces the trust in providers and make a negative effect on the receiving treatment and the necessary follow-up [37]. In a qualitative study, women stated that they did not receive the necessary information in their treatment procedure and did not participate in making the decisions and so their dignity was not conserved, and so they decided to discontinue the treatment [38].

Providing professional care and standard services was found to be another criterion for providing dignity-based infertility treatment services. In the International Conference of Population and Development in Cairo in 1994, prevention and appropriate treatment of infertility was emphasized as an important health priority, but unfortunately, it is still a great dilemma in the health systems of many countries [39]. Unaffordable treatment options, inadequate funds, and experts especially in public centers are the most common problems in providing infertility treatment services [40]. In a qualitative study, the participants mentioned the importance of accessible, continuous, and coordinate care with minimized cost for providing patient-centered care in infertility standard treatment services [41]. The current infertility policies cause an inappropriate distribution of public and private centers [42]. Financial problems in providing medicine and infertility treatment and inappropriate coverage of insurance requirements are to be solved in providing infertility care services [43].

The finding showed that considering the cultural and social aspects of infertility is an important dimension for providing dignity-based services. A few strategies such as attention to the common beliefs of society, modification of misbeliefs and attitudes, and recruiting eligible personnel were suggested. In Iranian and Islamic culture children are considered as a gift. Besides, it is difficult for the community to understand couples' life without kids, and childlessness is considered

stigmatization which is called "stove blindness" [44,45]. In a qualitative study using social constructivism theory, Hasanpour et al demonstrated that infertile couples face several mental and psychological problems that are created by society and are the product of the complex interaction of social relations, expectations, women's needs, the definition of infertility by society, and the view of society towards infertile people [46]. The necessity for educating the family and community for making proper beliefs and behaviors about infertility and infertile couples is emphasized in several studies [47–51]. Consistent with our findings Sami and Saeed Ali showed that women prefer to be treated by the same sex care providers and local personnel who are familiar with the infertility-related culture in that region because they can understand the infertility cultural problems [38].

Dignity-based models in other reproductive health contexts emphasize autonomy, respect, and support for individuals' reproductive choices. These models can be applied to various aspects of reproductive health, including family planning, contraception, pregnancy, and related service and its findings are similar to ours in this study, and integrating these findings will be effective in improving service delivery [36,38,44,47,49].

## Limitation and strength

The main limitation of this study similar to any other qualitative study, is the non-generalizability of the findings. The strength of the study is the explanation of the concept of dignity-based care in infertility treatment services using the perceptions of infertile couples and the key informants with diverse characteristics.

## Conclusion

Dignity-based care in infertility treatment services means "conserving dignity (respect, privacy, effective communication) in providing care services; making the information accessible and obtaining informed consent for care procedure; providing professional care and standard services; and considering cultural and social aspects of infertility." This concept can be used in future policy-making and planning, and appropriate support should be taken into account to improve the quality of infertility treatment services.

## Acknowledgments

The authors express their gratitude to women participating in the research, employees of infertility centers and all the researchers whose articles were analyzed, and the Nursing and Midwifery Faculty of the University of Medical Sciences for their support and contribution to this work.

## Author contributions

**Conceptualization:** Zahra Kiani, Masoumeh Simbar.

**Investigation:** Soheila Nazarpour.

**Methodology:** Zahra Kiani, Masoumeh Simbar, Vida Ghasemi.

**Writing – original draft:** Zahra Kiani, Masoumeh Simbar.

**Writing – review & editing:** Masoumeh Simbar.

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
