## [Decision Letter · Decision Letter 0]

18 Jun 2025

Dear Dr.  Simbar,

We look forward to receiving your revised manuscript.

Kind regards,

Sidrah Nausheen, FCPS

Academic Editor

PLOS ONE

Journal Requirements:

2. In this instance it seems there may be acceptable restrictions in place that prevent the public sharing of your minimal data. However, in line with our goal of ensuring long-term data availability to all interested researchers, PLOS’ Data Policy states that authors cannot be the sole named individuals responsible for ensuring data access (http://journals.plos.org/plosone/s/data-availability#loc-acceptable-data-sharing-methods).

Reviewers' comments:

Reviewer's Responses to Questions

**Comments to the Author**

1. Is the manuscript technically sound, and do the data support the conclusions?

Reviewer #1: Yes

Reviewer #2: Yes

2. Has the statistical analysis been performed appropriately and rigorously?

Reviewer #1: Yes

Reviewer #2: Yes

3. Have the authors made all data underlying the findings in their manuscript fully available?

Reviewer #1: No

Reviewer #2: Yes

4. Is the manuscript presented in an intelligible fashion and written in standard English?

Reviewer #1: Yes

Reviewer #2: Yes

Reviewer #1: few Typo Errors

Data sheets were written, not able to provide? If journal need Audio with study Code. It should be provided

Dignity based care/ themes generated in this qualitative research, how it is different from General care required by any health seeker including confidentiality. Social stigma can be different only.

Reviewer #2: This manuscript presents a qualitative study exploring the concept and dimensions of dignity-based care in infertility treatment services in Iran. Given the complex social, psychological, and ethical challenges surrounding infertility, particularly in culturally sensitive contexts, this research fills an important gap in literature by offering a multidimensional framework for understanding how dignity can be preserved in infertility care.

The study is well-structured, clearly written, and includes an extensive review of related literature. The authors employ a solid qualitative methodology grounded in content analysis, and the findings are categorized meaningfully into four central themes. The paper makes a strong case for policy-level interventions and service reforms that consider patients’ dignity across psychological, cultural, and systemic dimensions.

Methodology:

The methodological rigor of this qualitative study is commendable. The authors used conventional content analysis based on Graneheim and Lundman’s framework and adhered to Lincoln and Guba’s criteria for trustworthiness, which greatly enhances the credibility, dependability, confirmability, and transferability of the findings.

However, there are areas where methodological transparency could be improved:

• A brief table or explanation of the interview guide questions (even in supplementary material) would be helpful for reproducibility.

• While the authors mention “data saturation,” more detail on how this was operationalized (e.g., at which point no new themes emerged) would strengthen the claims.

• Clarification on how male and female narratives were comparatively analyzed could further enrich the gendered understanding of dignity in care.

Findings:

The manuscript presents four main categories that define dignity-based care in infertility services:

1. Conserving dignity in providing care

2. Access to information and informed consent

3. Providing professional care and standard services

4. Considering cultural and social aspects of infertility

These categories are richly supported by direct quotations, which provide authenticity and depth.

However, to improve the findings section:

• The presentation of Table 3 (codes, subcategories, categories) is dense and could benefit from better visual formatting or thematic grouping for readability.

Discussion:

The discussion is thorough and reflective, connecting the study’s findings with broader theoretical, ethical, and policy discourses on patient dignity and infertility treatment. The authors reference a range of philosophical and practical literature, offering a thoughtful framework to interpret their results.

Suggestions for enhancement:

• The integration of international best practices or dignity-based models in other reproductive health contexts could offer useful comparative insights.

Clarity, Style, and Organization:

The manuscript is generally clear and well-organized. The narrative flows logically from introduction to conclusion. Academic tone and language are appropriate, and the use of first-hand participant quotes adds richness.

A few editorial recommendations:

11categoires and 4 categories.....should be 11 sub categories and 4 categories..... in table 3

• Use consistent terminology: “dignity-based care,” “dignity-oriented care,” and “respectful care” appear somewhat interchangeably; a brief clarification of these terms early on would help.

The manuscript is a valuable, original, and well-executed contribution to the literature on infertility care and reproductive health ethics. Minor clarifications and edits, particularly in formatting, methodological detailing, and some stylistic improvements, will further enhance the quality and readability of the work.

**Do you want your identity to be public for this peer review?** For information about this choice, including consent withdrawal, please see our Privacy Policy

Reviewer #1: **Yes: ** Dr Farheen Yousuf. Assistant Professor Aga Khan University

Reviewer #2: **Yes: ** Dr Iffat Ahmed

---

## [Author Response · Author response to Decision Letter 1]

2 Jul 2025

Dear Editor:

Dignity-based care and infertility treatment: A qualitative study (PONE-D-25-11046)

Thanks to the valuable comments of the referees and the editor. The research team tried to prepare the answers and color the correct text in red.

Is the manuscript technically sound, and do the data support the conclusion?1.

Reviewer #1: Yes

Reviewer #2: Yes

R-Thanks

2. Has the statistical analysis been performed appropriately and rigorously?

Reviewer #1: Yes

Reviewer #2: Yes

R-Thanks

3. Have the authors made all data underlying the findings in their manuscript fully available?

Reviewer #1: No

Reviewer #2: Yes

R-Thanks

We tried to include all available information in the methods section to maintain confidentiality.

4. Is the manuscript presented in an intelligible fashion and written in standard English?

Reviewer #1: Yes

Reviewer #2: Yes

R-Thanks

5. Review Comments to the Author

Reviewer #1: few Typo Errors

Data sheets were written, not able to provide? If journal need Audio with study Code. It should be provided.

R- It is Corrected.

We tried to include all available information in the methods section to maintain confidentiality,

Dignity based care/ themes generated in this qualitative research, how it is different from General care required by any health seeker including confidentiality. Social stigma can be different only

We transcribed all interviews with women, men, and key informants and put similar codes together to arrive at the concept of a dignity-centered care unit.

Reviewer #2: This manuscript presents a qualitative study exploring the concept and dimensions of dignity-based care in infertility treatment services in Iran. Given the complex social, psychological, and ethical challenges surrounding infertility, particularly in culturally sensitive contexts, this research fills an important gap in literature by offering a multidimensional framework for understanding how dignity can be preserved in infertility care.

R-Thanks for the comment

The study is well-structured, clearly written, and includes an extensive review of related literature. The authors employ a solid qualitative methodology grounded in content analysis, and the findings are categorized meaningfully into four central themes. The paper makes a strong case for policy-level interventions and service reforms that consider patients’ dignity across psychological, cultural, and systemic dimensions

R-Thanks for the comment

Methodology:

The methodological rigor of this qualitative study is commendable. The authors used conventional content analysis based on Graneheim and Lundman’s framework and adhered to Lincoln and Guba’s criteria for trustworthiness, which greatly enhances the credibility, dependability, confirmability, and transferability of the findings.

R-Thanks for the comment

However, there are areas where methodological transparency could be improved:

• A brief table or explanation of the interview guide questions (even in supplementary material) would be helpful for reproducibility

R-Thanks for the comment. It is Corrected.

While the authors mention “data saturation,” more detail on how this was operationalized (e.g., at which point no new themes emerged) would strengthen the claims.

R-Thanks for the comment. It is Corrected.

• Clarification on how male and female narratives were comparatively analyzed could further enrich the gendered understanding of dignity in care.

We transcribed all interviews with women, men, and key informants and put similar codes together to arrive at the concept of a dignity-centered care unit.

Findings:

The manuscript presents four main categories that define dignity-based care in infertility services:

1. Conserving dignity in providing care

2. Access to information and informed consent

3. Providing professional care and standard services

4. Considering cultural and social aspects of infertility

These categories are richly supported by direct quotations, which provide authenticity and depth.

However, to improve the findings section:

• The presentation of Table 3 (codes, subcategories, categories) is dense and could benefit from better visual formatting or thematic grouping for readability.

We have provided more information in Figure 1 to make it more visual.

Discussion:

The discussion is thorough and reflective, connecting the study’s findings with broader theoretical, ethical, and policy discourses on patient dignity and infertility treatment. The authors reference a range of philosophical and practical literature, offering a thoughtful framework to interpret their results.

R-Thanks for the comment

Suggestions for enhancement:

• The integration of international best practices or dignity-based models in other reproductive health contexts could offer useful comparative insights.

R-Thanks for the comment. It is Corrected.

Clarity, Style, and Organization:

The manuscript is generally clear and well-organized. The narrative flows logically from introduction to conclusion. Academic tone and language are appropriate, and the use of first-hand participant quotes adds richness.

R-Thanks for the comment.

A few editorial recommendations:

11categoires and 4 categories.....should be 11 sub categories and 4 categories..... in table 3

R-Thanks for the comment. It is Corrected.

• Use consistent terminology: “dignity-based care,” “dignity-oriented care,” and “respectful care” appear somewhat interchangeably; a brief clarification of these terms early on would help.

R-Thanks for the comment. It is Corrected.

The manuscript is a valuable, original, and well-executed contribution to the literature on infertility care and reproductive health ethics. Minor clarifications and edits, particularly in formatting, methodological detailing, and some stylistic improvements, will further enhance the quality and readability of the work.

R-Thanks for the comment.

Other comment in text:

If the participant doesn’t know Dignity care, did you provide information?

The first and second questions are contradictory. AS per the objective of the study, to know about the expectations of patients while getting care

R-Thanks for the comment. We tried to fix this problem by adding more questions to the manuscript.

Rephrase for more clarity

R-Thanks for the comment. It is Corrected.

Do you mean wife or husband as the participant is a female?

R-Thanks for the comment. It is Corrected.

Kindly rephrase it. The immediate concept came as same sex couples.

R-Thanks for the comment. It is Corrected.

---

## [Decision Letter · Decision Letter 1]

25 Sep 2025

Dignity-based care and infertility treatment: A qualitative study

PONE-D-25-11046R1

Dear Dr. Simbar,

We’re pleased to inform you that your manuscript has been judged scientifically suitable for publication and will be formally accepted for publication once it meets all outstanding technical requirements.

Kind regards,

Stefan Schlatt

Academic Editor

PLOS ONE

Additional Editor Comments (optional):

Reviewers' comments:

Reviewer's Responses to Questions

**Comments to the Author**

Reviewer #1: All comments have been addressed

2. Is the manuscript technically sound, and do the data support the conclusions?

Reviewer #1: Yes

3. Has the statistical analysis been performed appropriately and rigorously?

Reviewer #1: Yes

4. Have the authors made all data underlying the findings in their manuscript fully available?

Reviewer #1: Yes

5. Is the manuscript presented in an intelligible fashion and written in standard English?

Reviewer #1: Yes

Reviewer #1: The author addressed all concerns raised and answer appropriately. The flow of writing is good. All parametersare fulfilled.

**Do you want your identity to be public for this peer review?** For information about this choice, including consent withdrawal, please see our Privacy Policy

Reviewer #1: No

---

## [Editor Report · Acceptance letter]

PONE-D-25-11046R1

PLOS ONE

Dear Dr. Simbar,

I'm pleased to inform you that your manuscript has been deemed suitable for publication in PLOS ONE. Congratulations! Your manuscript is now being handed over to our production team.

Kind regards,

on behalf of

Dr. Stefan Schlatt

Academic Editor

PLOS ONE